# Testosterone pulses paired with a location induce a place preference to the nest of a monogamous mouse under field conditions

**Radmila Petric[1,2]\*, Matina Kalcounis-Rueppell[2,3], Catherine A Marler[4]**

[1]Institute for the Environment, University of North Carolina at Chapel Hill, Chapel Hill, United States; [2]Biology, University of North Carolina at Greensboro, Greensboro, United States; [3]Biological Sciences, University of Alberta, Edmonton, Canada; [4]Psychology, University of Wisconsin-Madison, Madison, United States

**Abstract** Changing social environments such as the birth of young or aggressive encounters present a need to adjust behavior. Previous research examined how long-term changes in steroid hormones mediate these adjustments. We tested the novel concept that the rewarding effects of transient testosterone pulses (T-pulses) in males after social encounters alter their spatial distribution on a territory. In free-living monogamous California mice (*Peromyscus californicus*), males administered three T-injections at the nest spent more time at the nest than males treated with placebo injections. This mimics T-induced place preferences in the laboratory. Female mates of T-treated males spent less time at the nest but the pair produced more vocalizations and call types than controls. Traditionally, transient T-changes were thought to have transient behavioral effects. Our work demonstrates that in the wild, when T-pulses occur in a salient context such as a territory, the behavioral effects last days after T-levels return to baseline.

**\*For correspondence:**
r_petric@uncg.edu

**Competing interest:** The authors declare that no competing interests exist.

## Editor's evaluation

Manipulations of sex hormones in animals in ecologically relevant environments usually involve long-term manipulations using chronic implants or injections of esterified steroids with longer half-lives than the endogenous hormones. This has been done in line with the prevailing idea of the long-lasting effects of steroids mediated by the transcritpional actions of their liganded receptors. The specific novelty of this study lies in the transiency of hormone availability (testosterone's half-life is about 2 hours). This might suggest that the observed effects depend on a mode of action different from the mode of action during chronic sex hormone exposure. It should also be noted that any study in natural settings is significantly more difficult to perform than in the lab. However, as all brain/hormonal functions evolved in natural environments, these studies are absolutely crucial to understanding the function of the respective systems.

## Introduction

Animals frequently adjust their allocation of time as they move through various life-history stages and meet different social challenges; we ask what mechanisms alter preferences for physical locations in the wild? One mechanism for altering the approach to a stimulus is through rewarding or reinforcing neural processes (*Glickman and Schiff, 1967*) such as the repeated linkage between the rewarding properties of a pulse of testosterone (T) and the presence of a stimulus. We proposed that, as in the

laboratory (e.g. *Zhao and Marler, 2014*; *Zhao and Marler, 2016*), natural male T-pulses occurring after social interactions with males or females would function differently from long-term implants in the field (*Fusani, 2008*; *Goymann et al., 2015*; *Ketterson et al., 1992*; *Marler and Moore, 1989*; *Nyby, 2008*) by creating a preference for a specific location within a territory in the wild. One scenario for explaining a possible difference between T-implants and T-pulses is that while T-implants function through classical androgen and estrogen receptors (after conversion to estrogen), the rewarding, possibly more rapid effects, of T can occur through 'nongenomic' actions of androgens (*Sato et al., 2010*). T would then act as an internal reward (*Gleason et al., 2009*) or reinforcing stimulus such that when released naturally or through an injection, increase approach to the physical location in which the T-pulse was experienced, as occurs under laboratory conditions in rodents (e.g. *Zhao and Marler, 2014*). The reinforcing effects occur via activation of the neural internal reward system (e.g. *Bell and Sisk, 2013*). This effect has potentially broad reaching applications because male T-pulses are released in response to different social interactions across a variety of species including humans (*Gleason et al., 2009*). In the case of a biparental species, T-release near the nest may provide a mechanism for increasing a male's attendance at the nest for at least several days, as suggested by the results of a laboratory study (*Zhao and Marler, 2014*) using classical conditioned place preference (CPP) tests (*Arnedo et al., 2000*; *Frye et al., 2001*). We explore the hormone T as a stimulus that has rewarding/reinforcing effects (*Arnedo et al., 2000*; *Frye et al., 2001*; *Zhao and Marler, 2014*; *Zhao and Marler, 2016*; *Zhao et al., 2019*; *Zhao et al., 2020*), albeit a weak effect compared to drugs of abuse (*Roozen et al., 2004*), in the wild with many relevant, competing stimuli from the natural surrounding environment.

A classic formalized hypothesis related to T-release in male-male interactions is the 'Challenge Hypothesis' stating that male-male encounters induce increases in T in response to challenges from other males (*Wingfield et al., 1990*). In a series of laboratory studies in this monogamous, biparental, and highly territorial California mouse (*Peromyscus californicus*), we found that T-pulse release occurs after male-male aggressive encounters that influence future behavior under laboratory conditions (*Gleason et al., 2009*; *Fuxjager et al., 2011*; *Marler et al., 2005*; *Oyegbile and Marler, 2005*; *Trainor et al., 2004*; *Zhao and Marler, 2014*; *Zhao and Marler, 2016*). Plasticity in the rewarding nature of these T-pulses has been discovered in this monogamous species, such that the formation of CPPs can be dependent on the familiarity of the environment and the pair-bond status (*Zhao and Marler, 2014*; *Zhao and Marler, 2016*). For example, in pair-bonded California mice, T-pulses induce CPPs in familiar but not unfamiliar environments (*Zhao and Marler, 2014*; *Zhao and Marler, 2016*). Specifically, a male receiving a T-injection in the middle chamber where he has a nest and is a resident (increased ability to win a male-male encounter after 24 hr residency) and with his mate temporarily removed (no pups) will form a CPP to the nest chamber but not the less familiar side chambers (*Zhao and Marler, 2014*). Interestingly, the opposite is true for sexually naïve males, in which T-pulses induce CPPs in unfamiliar side chambers, but not in familiar environments (*Zhao and Marler, 2014*; *Zhao and Marler, 2016*). Therefore, the function of these T-pulses is dependent on social interactions and location. Significantly, T-release occurs in response to female stimuli as well (Zhao and Marler, unpublished data). Female stimuli are known to evoke both T-pulses (*Nyby, 2008*; Zhao and Marler unpublished data) and CPPs from males (e.g. *Bell et al., 2010*; *Meisel and Joppa, 1994*). Interestingly, T and its releasing hormone gonadotropin releasing hormone (*George et al., 2021*) can also have positive effects on paternal behavior in some species (reviewed by *Guoynes and Marler, 2020*).

T-pulses modulate other behaviors such as vocalizations (*Pultorak et al., 2015*; *Remage-Healey and Bass, 2006*), which can affect aspects of sexual selection. Within minutes of a T-pulse in Gulf toadfish (*Opsanus beta*) and plainfin midshipman fish (*Porichthys notatus*), males increased call rate and duration of calls which females prefer (*Remage-Healey and Bass, 2004*; *Remage-Healey and Bass, 2006*). Male California mice administered a single T-pulse and placed in the presence of a novel female decreased production of calls associated with courtship in pair-bonded but not unpaired males in the laboratory (*Pultorak et al., 2015*). This finding indicates that in California mice, bonding likely induces a neural change that alters the response to T-pulses and reduces vocal courtship responsiveness to unfamiliar females (*Pultorak et al., 2015*). T-pulses also have long-term effects on call production in California mice, such that days after multiple T-pulse injections in the field, males produced more call types with a nonsignificant trend to produce more ultrasonic vocalizations (USVs) (*Timonin et al., 2018*).

We hypothesized that, in the wild, T-pulses would reinforce behaviors in the area where the social experiences induced T-pulses through the formation of CPPs that would, in turn, alter associated social behavior. Here, we tested three predictions: (1) pair-bonded males receiving T-injections at the nest would spend more time at the nest; (2) females would adjust for the increased time that her T-injected mate spent at the nest by decreasing her time at the nest and allocating more time to activities away from the nest (based on *Trainor and Marler, 2001*); (3) T-pulses would induce changes in type and number of USVs produced as part of both the direct effects of T on behavior and the indirect

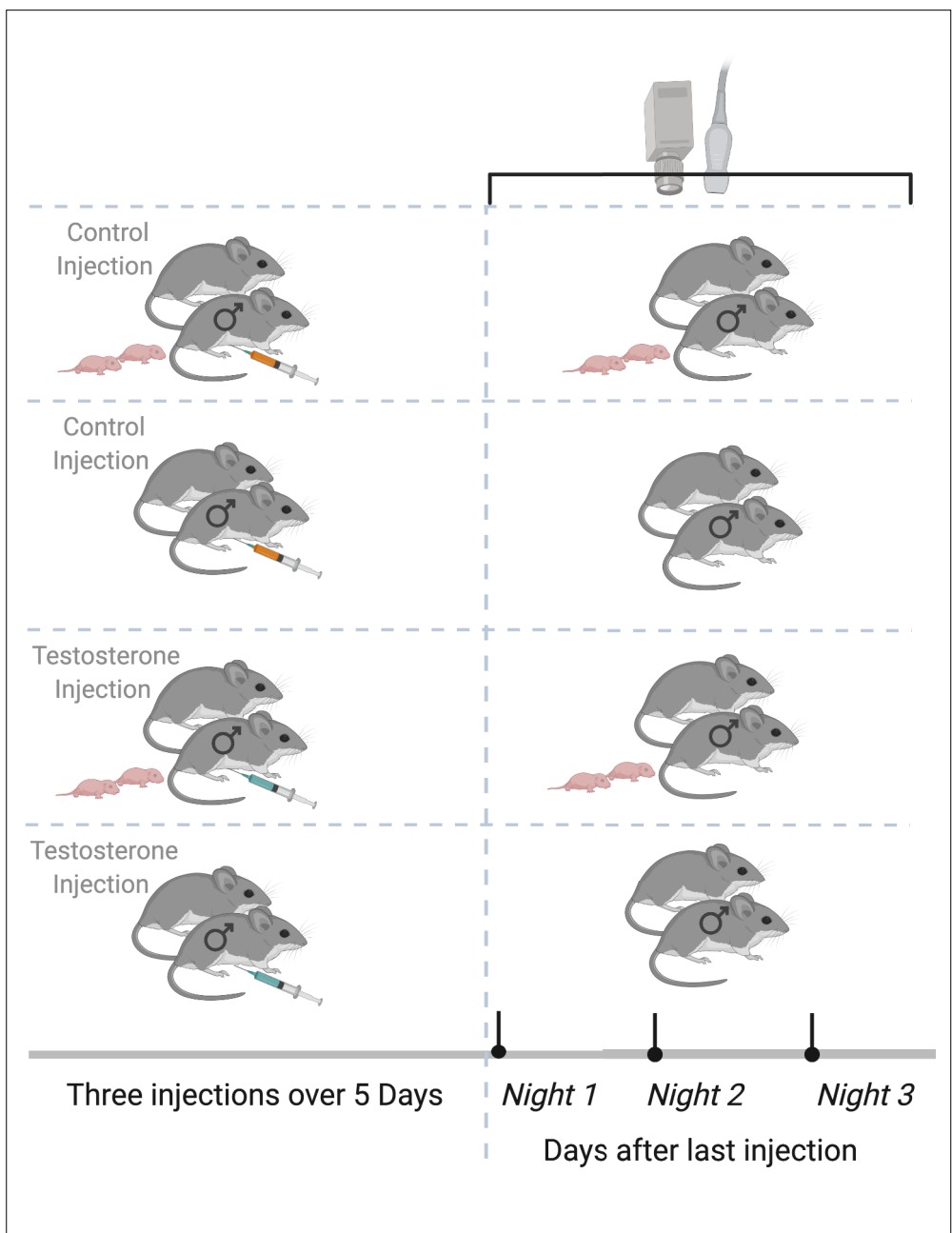

**Figure 1.** Experimental design. Paired male California mice (*Peromyscus californicus*) with and without pups were randomly assigned to receive three subcutaneous injections over five nights of either testosterone (**T**) or saline/control (**C**). After the third and last injection, we deployed the remote sensing equipment (automated radio telemetry, audio recording, and thermal imaging) to record individual behaviors for three consecutive nights. Data were collected from California mice at the Hastings Natural History Reserve in 2015. Created with https://biorender.com/.

effects on the pairs' social adjustment to the altered time allocation to a specific location (*Timonin et al., 2018*).

We tested our hypothesis in the well-studied monogamous and territorial California mouse by administering three T-pulses to paired males at the nest site (*Figure 1*; see Materials and methods for details). In this species, males balance their time between behaviors such as mate attendance, offspring care, and territory defense (*Gubernick and Alberts, 1987*; *Gubernick et al., 1993*; *Gubernick and Teferi, 2000*). In the laboratory and the wild, California mouse adults frequently produce USVs. In the wild, sustained vocalizations (SVs) and barks are reliably recorded (*Briggs and Kalcounis-Rueppell, 2011*; *Kalcounis-Rueppell et al., 2006*; *Kalcounis-Rueppell et al., 2010*; *Kalcounis-Rueppell et al., 2018*; *Timonin et al., 2018*). SVs are the most common call type recorded in the field as single calls or bouts of multiple calls that are categorized based on the number of calls in a bout (1SV, 2SV, 3SV, 4SV; *Kalcounis-Rueppell et al., 2018*). SVs are long, low modulation calls with harmonics that may serve as both long-distance contact vocalizations (*Briggs and Kalcounis-Rueppell, 2011*) and to convey aggression when in a shortened form (*Rieger and Marler, 2018*). Free-living California mice maintain strict territories (*Ribble and Salvioni, 1990*), therefore, social interactions at the nest occur primarily between pair members and include production of SVs as is consistent with production of SVs between pairs in the laboratory (*Pultorak et al., 2018*; *Pultorak et al., 2017*). Thus, the monogamous reproductive system of the California mouse and their known time management and production of vocalizations contribute to a compelling system for assessing behavioral responses to T-pulses and

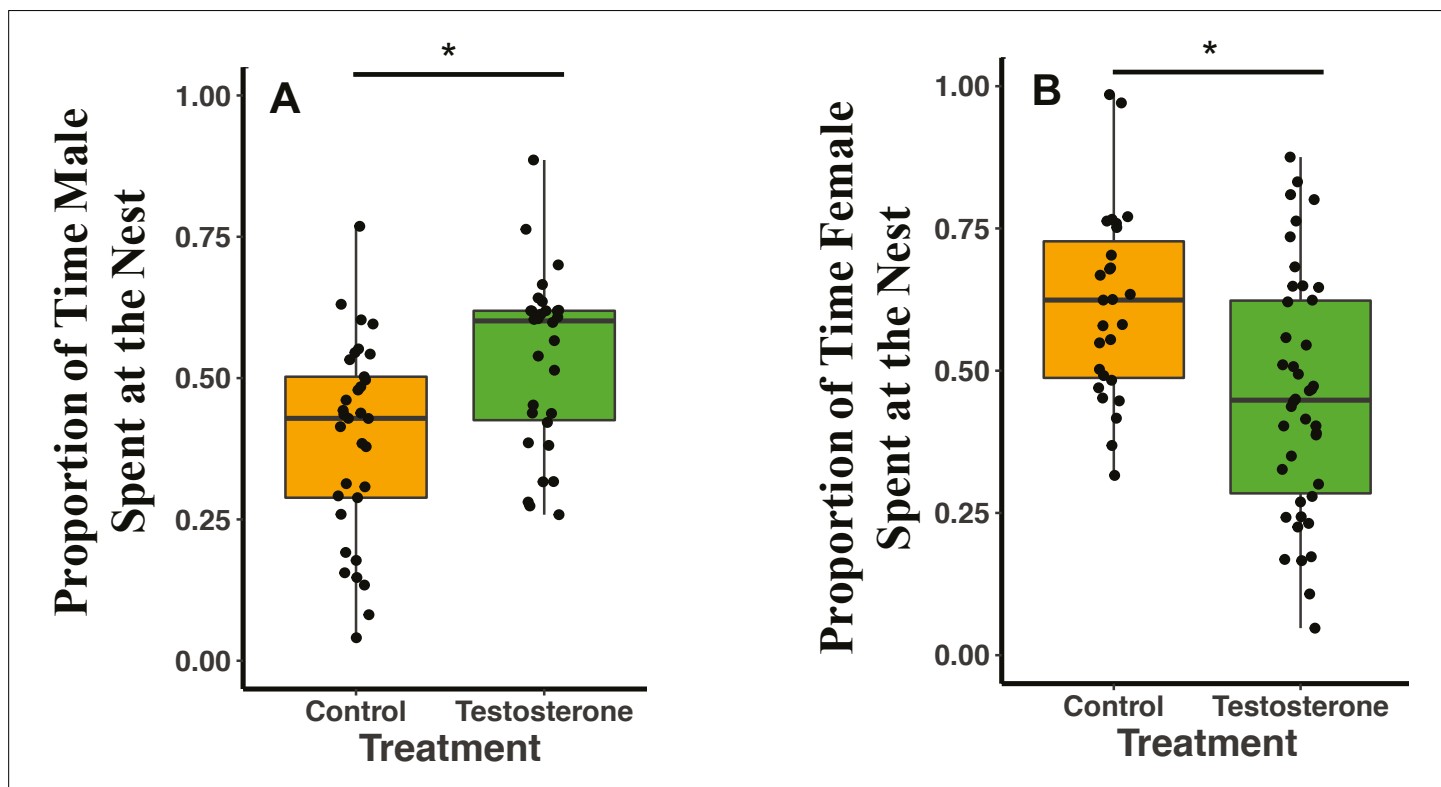

**Figure 2.** Median and quantiles of proportion of time at the nest by treatment type (C or T). (**A**) Proportion of time at the nest for males (T: n = 10 and C: n = 11). T-males spent 14% *more* time at the nest than C-males (GLMM Estimate 0.14 ± 0.05, p = 0.02). (**B**) Proportion of time at the nest for females (T: n = 14 and C: n = 9). T-females spent 15.8% *less* time at the nest than C-females (GLMM Estimate –0.16 ± 0.06, p = 0.02). A single dot represents the observations from one individual on a single night. For each individual, there are therefore three dots in the figure representing three nights (reflecting our GLMM analysis). There is no loss of statistical significance if data are analyzed with individual averages instead of repeated measures (see Appendix 1). *Source data 1*.

The online version of this article includes the following figure supplement(s) for figure 2:

**Figure supplement 1.** Median and quantiles of male time at the nest by treatment type and by presence of pups.

**Figure supplement 2.** Median and quantiles of female time at the nest by male treatment type and by presence of pups.

the establishment of male T-induced CPP in the field to alter the amount of time that males spend at the nest.

## Results

### Time at the nest

Overall, T-males spent 14% more time at the nest (defined as within 2 m of the nest) than C-males (GLMM Estimate 0.14 ± 0.05, p = 0.02; *Figure 2A*; see also *Supplementary file 1A*). Females were not subjected to T-injections, but we examined their responses to their T-injected mates. T-females spent 15% less time at the nest than C-females (GLMM Estimate –0.16 ± 0.06, p = 0.02; *Figure 2B*; *Supplementary file 1B*). T- and C-females spent more time at the nest on night three of recording compared to night one of recording (night three GLMM Estimate 0.10 ± 0.04, p < 0.02; *Supplementary file 1B*). T-females spent 13% more time at the nest on night three than night one and C-females spent 6% more time on night three than night one (*Supplementary file 1B*). Female time at the nest was negatively influenced by male T-injections (T: GLMM Estimate –0.15 ± 0.07, p = 0.04; *Supplementary file 1C*) and by male time at the nest (Time at the Nest: GLMM Estimate 0.36 ± 0.17, p = 0.04; *Supplementary file 1C*). T-females spent 5% less time at the nest than their mates, whereas, C-females spent 18% more time at the nest than their mates (*Supplementary file 1C*).

Males and females spent more time at the nest when there were pups (male time at the nest and pup presences GLMM Estimate –0.21 ± 0.04, p < 0.00; female time at the nest and pup presences GLMM Estimate 0.19 ± 0.06, p < 0.00), however, sample sizes were too small to statistically compare both pup presence and treatment type in one model. Data are shown in (a) *Figure 2—figure supplements 1 and 2* and (b) *Supplementary file 1A*.

Proportion of male time at the nest was not statistically influenced by season (GLMM Estimate –0.09 ± 0.06, p = 0.17), body mass (GLMM Estimate –0.01 ± 0.01, p = 0.51), total nights needed to administer all three injections (GLMM Estimate –0.09 ± 0.08, p = 0.26) or recording night (comparing night one to night two GLMM Estimate 0.04 ± 0.04, p = 0.39; or night three GLMM Estimate 0.07 ± 0.04, p = 0.11 *Supplementary file 1A*). In contrast to males, female time at the nest appears to be partially influenced by other factors. T-females spent 15.6% less time in the nest during spring than fall (spring GLMM Estimate –0.15 ± 0.06, p = 0.02; *Supplementary file 1B*). C-females spent 10.3%

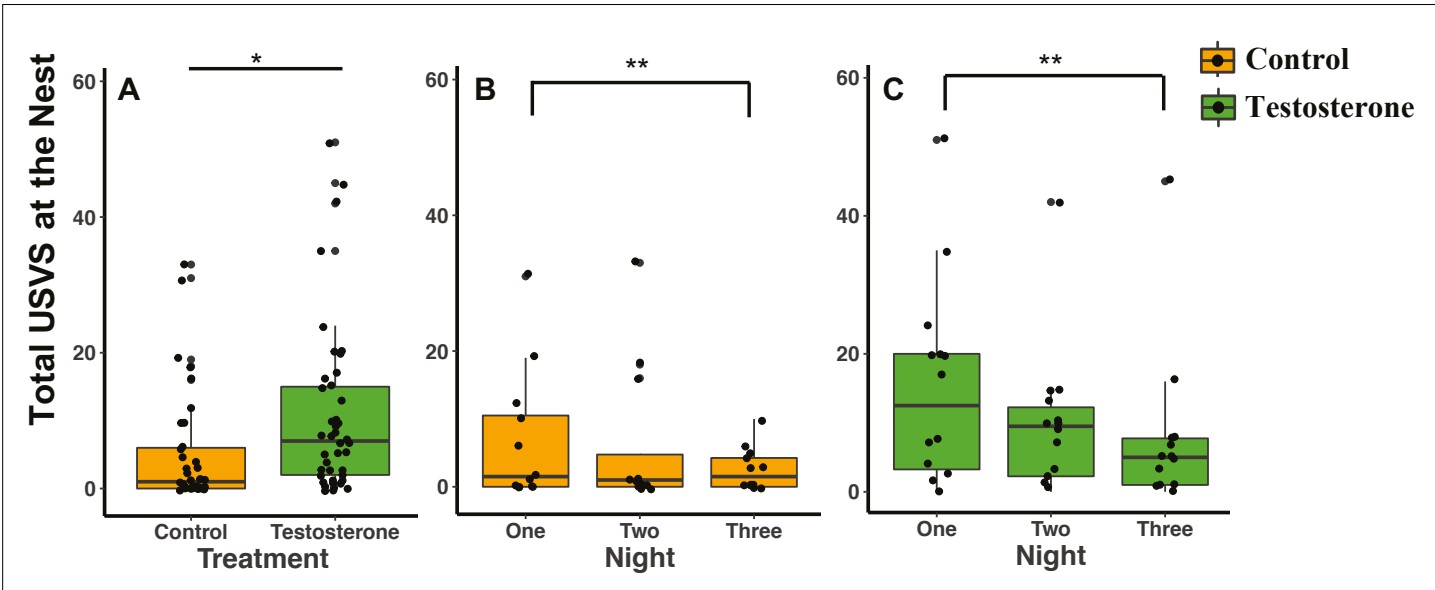

**Figure 3.** Median and quantiles of ultrasonic vocalizations (USVs) produced at the nest based on treatment and the three nights following the last injection. (**A**) Pairs produced more total USVs at T-nests (n = 14 dyads) than C-nests (n = 12 dyads) (GLMM Estimate 0.87 ± 0.40, p = 0.04). (**B and C**) The number of total USVs produced by C-pairs and T-pairs decreased from night one to night three (GLMM Estimate –0.76 ± 0.26, p < 0.01). In figure (**A**) only, a single dot represents the observations from one pair on a single night. In figure (**A**), there are therefore three dots per pair representing each of the three nights (reflecting our repeated measures GLMM analysis). Figures (**B**) and (**C**) are broken down by treatment and by night and therefore each pair is represented by one dot per night. *Source data 1.*

less time at the nest during spring than fall (*Supplementary file 1B*). Female time at the nest was not, however, influenced by body mass (GLMM Estimate 0.01 ± 0.01, p = 0.24) or mass difference between the female and the male (GLMM Estimate 0.01 ± 0.01, p = 0.17).

## Total USVs

We recorded a total of 549 total USVs across the 26 nest sites (T USVs = 368, C USVs = 181). All call types (1-, 2-, 3-, 4-, 5-, 6SV, and barks) were recorded for the male and the female at both C- and T-nests. Of the 26 pairs, 22 contributed to an average of 23.87 ± 20.95 USVs per pair. While all recordings were made at the nest, we further assigned context based on the distance between the members of a pair (apart: >2 m; together: <1 m; intermediate: 1–2 m apart) to 385 USVs. More USVs were produced when two mice were >2 m apart ($\chi 2$ = 9.99, df = 2, p = 0.01). When analyzed by distance, we found that 157 USVs were produced when a mouse was >2 m (T USVs = 101, C USVs = 56), 119 USVs were produced when the mouse was <1 m away from another mouse (T USVs = 94, C USVs = 25), and 109 USVs were produced when the mouse was 1–2 m away from another mouse (T USVs = 76, C USVs = 33).

When considering treatment type, T-pairs produced twice as many total USVs at the nest than C-pairs (GLMM Estimate 0.87 ± .40, p = 0.04; *Figure 3A*; *Supplementary file 1D*). Both C- and T-pairs produced twice as many USVs on recording night one than on recording night three (*Figure 3B and C*; *Supplementary file 1D*).

Independent of treatment, additional statistical analyses show that pairs also produced more USVs on night one than night three after the last injection (GLMM Estimate –0.76 ± 0.26, p = 0.01; *Figure 3C*; *Supplementary file 1D*), but there was no difference between night one and night two (GLMM Estimate –0.33 ± 0.26, p = 0.15; *Figure 3B*; *Supplementary file 1D*). The total number of USVs recorded was not influenced by pups (GLMM Estimate –0.48 ± 0.40, p = 0.25), season (GLMM Estimate –0.68 ± 0.40, p = 0.10), body mass (GLMM Estimate 0.01 ± 0.06, p = 0.92), or total nights

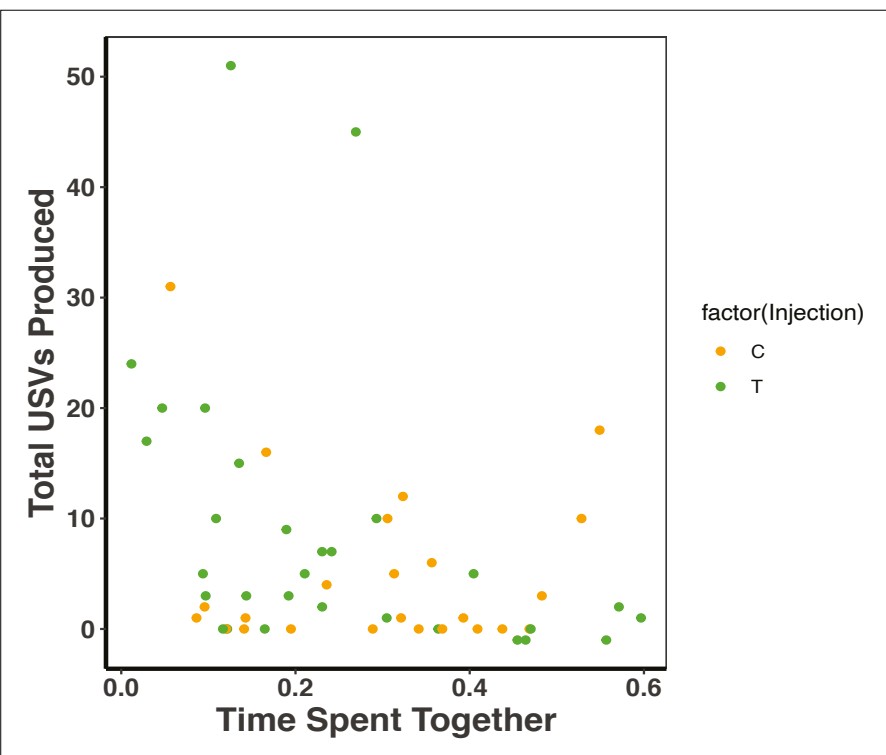

**Figure 4.** There was a negative association between total ultrasonic vocalizations (USVs) produced and time spent together as a dyad ($F_{2,51}$=20.68, $R^2$ = 0.12, p = 0.03). There was, however, no treatment effect on the total USVs produced and time spent together as a dyad ($F_{2,51}$ = 20.68, $R^2$ = 0.12, p = 0.37). A single dot represents the observations from one dyad on a single night (T: n = 10, C: n = 8 dyads). There are therefore three dots per dyad representing each of the three nights (reflecting our repeated measures GLMM analysis). *Source data 1*.

needed to administer all three injections (GLMM Estimate –0.85 ± 0.64, p = 0.20; *Supplementary file 1D*).

We further examined whether time spent together within pairs influenced USV production as a potential mediating factor for the association between treatment and USVs. When we combined treatments, there was a significant association between the time the pair spent together and USVs (time spent together $F_{2,51}$ = 20.68, $R^2$ = 0.12, p = 0.03; *Figure 4*) such that pairs that spent less time together produced more USVs (time spent together $F_{2,51}$ = 20.68, $R^2$ = 0.12, p = 0.03; treatment p = 0.37; *Figure 4*). We unfortunately could not tease apart the effect of T on USV number and time that the pair spent apart or together because of the logistical challenges of binning times related to animal movement. Therefore, the effect of T on USV production could still potentially be mediated by differences in time that pair mates of the different treatment groups were spending together.

## Call types

The number of each USV call type (1–6SVs) for both groups and each distance is included in *Supplementary file 1E*. As mentioned earlier, we exclude 5-, 6SVs, and barks from analyses because of small sample size. Based on distance alone, both male and female mice were more likely to produce SVs (all SV types combined) when the mate was >2 m from the nest than when located <1 m from the nest (GLM Estimate 0.52 ± 0.12 p < 0.01) and there was a nonsignificant trend for more USVs produced when the mice were 1–2 m apart than <1 m from the nest (GLM Estimate 0.22 ± 0.13, p = 0.09). There was a negative correlation between the number of USVs produced and female time at the nest (t = −1.96, df = 64, p = 0.05).

T-mice were more likely to produce SVs (all types combined) than C-mice (treatment GLM Estimate 0.72 ± 0.11 p < 0.01). More specifically, when all distances between pair members are combined, T-pairs produced proportionately more 4SVs than control pairs (W = 43, p = 0.03; *Supplementary file 1F*). There was no significant difference between treatments in any other proportion of call type produced (1-, 2-, 3SV; p > 0.137). We also have evidence that T-treatment influences specific SV types

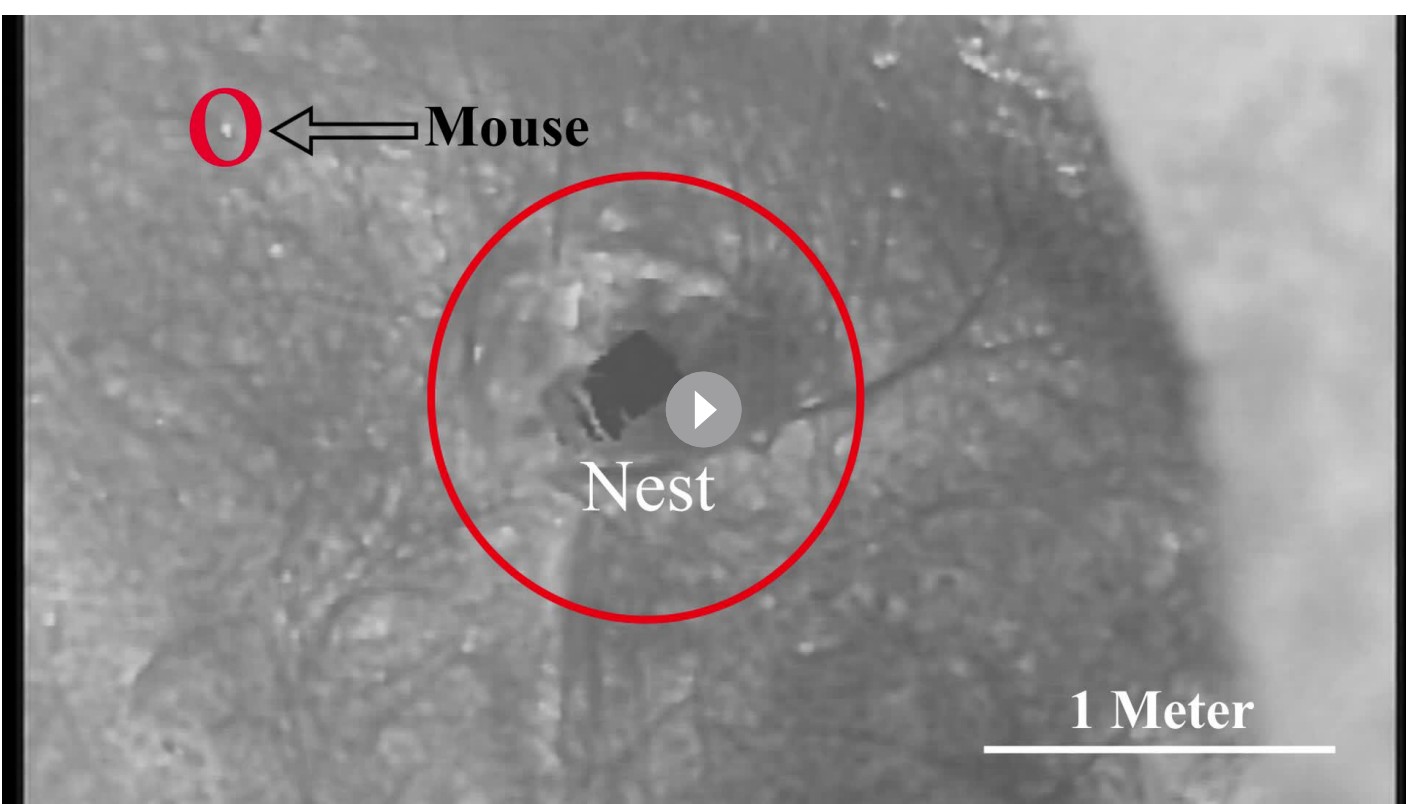

**Video 1.** Male California mouse at the nest.
https://elifesciences.org/articles/65820/figures#video1
**Video 1—source data 1.** Male California mouse at the nest.

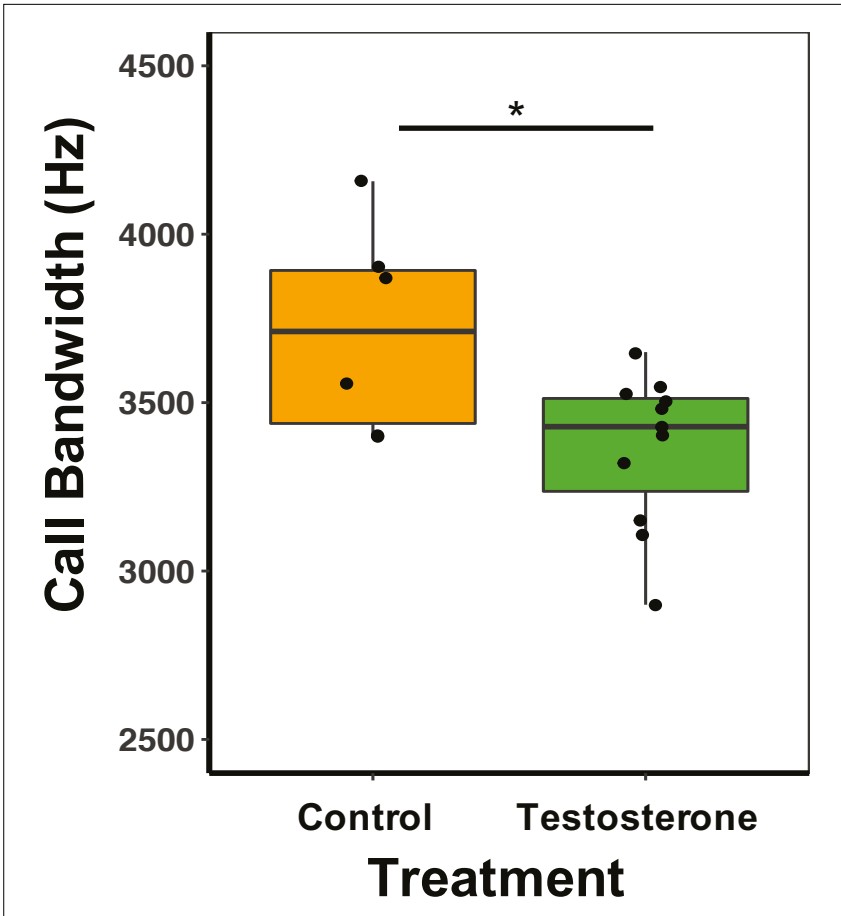

**Figure 5.** Median and quantiles of call bandwidth (Hz) for male mice. Bandwidth was measured in the first call in the sequence for 1, 2-, 3-, and 4SVs produced by males. T-males (n = 12) produced calls with a 11.25% smaller bandwidth than C-males (n = 6) (GLM Estimate –0.13 ± 0.01, p < 0.01). A single dot represents the average bandwidth value for an individual male. *Source data 2*.

The online version of this article includes the following figure supplement(s) for figure 5:

**Figure supplement 1.** PCA of the first call in the sequence for 1-, 2-, 3-, and 4SVs produced by males (T: n = 86 and C: n = 31).

when analyzed by distance from the nest based on proportion of total SVs. When > 2 m and 1–2 m apart (regardless of pup presence), T-mice were more likely to produce 1-, 2-, and 4SVs (1SV $\chi$2 = 9.95, df = 2, p < 0.01; 2SV $\chi$2 = 9.59, df = 2, p < 0.01; 4SV $\chi$2 = 9.48, df = 2, p < 0.01; *Video 1*) (again this was not controlled for time pair mates spent together) but not 3SVs (3SV $\chi$2 = 5.1, df = 2, p = 0.08). In C-mice there was no significant difference in the proportion of each SV type produced (1–4SVs) for any of the three distances (p > 0.15).

## Spectral and temporal characteristics of USVs

There was a treatment effect on call bandwidth, whereby T-males produced calls with a 11.25% smaller bandwidth than C-males (GLM Estimate –0.13 ± 0.01, p < 0.01; *Figure 5*; *Supplementary file 1G*). There was, however, no effect on other spectral or temporal characteristics of calls (*Supplementary file 1G*). There was no difference between treatment types in call duration (GLM Estimate –0.09 ± 0.12, p = 0.46) or PC1 score (GLM Estimate 0.77 ± 1.07, p = 0.48; *Supplementary file 1G*). For females, there was no significant difference between treatment type and any call characteristics, duration (GLM Estimate –0.09 ± 0.21, p = 0.68), bandwidth (GLM Estimate –0.11 ± 0.07, p = 0.88), or PC1 score (GLM Estimate 0.51 ± 1.02, p = 0.63; *Supplementary file 1F*).

## Discussion

A long-standing question in the field of behavioral neuroendocrinology asks what are the functions of short-term T-pulses that are induced by competitive, aggressive, and sexual interactions (e.g. *Ball and Balthazart, 2020*)? For the first time, we used a modified classic CPP paradigm to show that multiple T-pulses experienced in a specific location on a territory in the field can increase the amount of time that a male spends at that location; in this case increased time at his nest. In addition, males and females also spent more time at the nest when pups were present.

The nest is the most stable and salient location in a territory in the field. Moreover, the nest remains salient even without pups and when the mate is away; we therefore chose to start our series of studies with T-injections at the nest and monitored the nest and the area immediately surrounding it. The brief transient nature of the T-pulse allows it to be paired with specific stimuli in the field. The use of T-pulses via injections contrasts with long-lasting implants (and/or castrations) used in the past to examine effects of T on seasonal, long-term changes in behavior such as those associated with aggression, breeding, parental behavior, song, and spatial behavior in the field (e.g. *Chandler et al., 1997*; *Marler and Moore, 1988a*; *Marler and Moore, 1988b*; *Marler and Moore, 1989*; *Moore and Marler, 1987*; *Watson and Parr, 1981*; *Wingfield, 1984*). For example, T-implants cause increases in territorial patrolling in the mountain spiny lizard, *Sceloporus jarrovi* (*Marler and Moore, 1989*), larger home ranges and territories in both avian and lizard species (e.g. *Chandler et al., 1997*; *DeNardo and Sinervo, 1994*; *Watson and Parr, 1981*; *Wingfield, 1984*), decreased paternal care in the form of time at the nest (e.g. *Chandler et al., 1997*), and increased singing in birds (review by *Lynn, 2008*). Within rodents, long-term androgen manipulations in the laboratory can also alter vocalizations; for example, *Pasch et al., 2011a* found that castration resulted in fewer songs in male singing mice. With T-pulses in the current study, we found that males increased place preference for the nest while the female spent more time away from the nest. Moreover, the pair produced more calls primarily in the form of 4SVs. The similarity between the hormonal techniques is that both can influence vocalizations, although these likely have different functions. The increase in 4SVs in the current study likely function as contact calls between members of a pair in the monogamous and biparental California mice and because most occurred when the pair was apart. In contrast, the increase in songs of male singing mice in response to T appears to function directly in male-male aggression (*Pasch et al., 2011b*). A comparison of T-implants and T-pulses is needed within the same species to further this comparison, but it is expected that the formation of finely tuned CPPs is unique to T-pulses.

The comparison of mechanisms examining T-effects on behavior via baseline versus experience induced changes in T (mimicked by T-injections) also leads us to ask whether there are different mechanisms underlying the interaction between T and behavior. First, within California mice it is known that blocking T conversion to estradiol influences effects of baseline levels of T on aggression in the form of attack latency, but not the T-pulses that mimic experience-induced aggression; this suggests that baseline effects of T on aggression are related to estrogen receptors, and experience-induced effects are related to androgen receptors (*Trainor et al., 2004*). Importantly, the focus on T-implants also ignores the role of the rewarding aspects of T-pulses elicited by social interactions paired with environmental stimuli, such as location, that we argue can induce preferences for that location in the field; such an effect can result in more fine tuned location preferences within a territory based on social challenges that in this case appear to last days after the T-injections. The rewarding/reinforcing effects of T-pulses may well operate through other proposed cellular mechanisms; androgen reinforcement can act through membrane androgen receptors (*Wood, 2004*) and/or androgen metabolites (e.g. *Frye, 2007*; *Rosellini et al., 2001*). Such a mechanism has the potential to function more rapidly because it does not depend on direct gene transcription and a rapid effect of T, within minutes, remains to be tested for CPPs.

Two other broad concepts to emphasize are first that T-pulses may provide another neuroendocrine mechanism for allowing males to avoid the high costs of sustained T-levels characterized by decreased survivorship or condition (e.g. *Alonso-Alvarez et al., 2007*; *Buchanan et al., 2001*; *Dufty, 1989*; *Fuxjager et al., 2011*; *Sinervo et al., 2000*). Moreover, conditioning via T-pulses further supports the concept that T-pulses are another mechanism for altering androgen-influenced phenotypes, albeit probably more transient in nature (review by *Fuxjager and Schuppe, 2018*). Second, from a laboratory perspective, we found evidence consistent with the concept that the weak conditioning effects of T-pulses via CPPs can increase time allocation by a mammal to a location, the nest, within a territory

in the wild. The CPP behavioral paradigm is used extensively in laboratory studies for measuring the reinforcing and addictive nature of drugs and neurochemicals, but there is a gap in our understanding of the natural functions for these location preferences, including the relatively weak effects produced by T. This is important for understanding plasticity in the formation of rewarding/reinforcing effects of drugs, including those that result in location preferences.

## T and CPPs

By using T-injections, we mimicked the natural T-pulses that occur after male-male and male-female interactions in male California mice (*Marler et al., 2005*; *Oyegbile and Marler, 2005*; Zhao and Marler unpublished), as well as a number of other species including humans (recent reviews by *Maney et al., 2020*; *Moore et al., 2020*; *Wingfield et al., 2020*). In the context of CPPs, we previously found that these injections in the laboratory can alter both time spent in a location (*Zhao and Marler, 2014*; *Zhao and Marler, 2016*) and social behaviors (*Fuxjager et al., 2011*; *Pultorak et al., 2015*; *Trainor et al., 2004*; *Zhao and Marler, 2014*; *Zhao et al., 2019*; *Zhao et al., 2020*). Our results are consistent with laboratory observations in mice, rats, and hamsters showing that T-pulses have reinforcing/rewarding effects as described in the introduction (*Alexander et al., 1994*; *Arnedo et al., 2000*; *Wood, 2004*; *Zhao and Marler, 2014*; *Zhao and Marler, 2016*). It is of interest to note that the androgen-induced CPPs can be blocked by dopamine antagonists (*Becker and Marler, 2015*), further supporting the concept of reinforcing/reward functions (*Gleason et al., 2009*; *Marler et al., 2005*; *Packard et al., 1998*).

T-pulses in response to male-male social challenges is a defining hallmark of Wingfield's Challenge Hypothesis (*Wingfield et al., 1990*) but also occurs in males after male-female sexual interaction (*Gleason et al., 2009*). The importance of the male-female interaction in eliciting T-pulses across species has been highlighted by *Goymann et al., 2019*. Male mice and rats exposed to an estrous female or her olfactory cues show a preference for the location at which the sexual encounter occurred (*Camacho et al., 2004*; *Frye et al., 2001*; *Hughes et al., 1990*; *Mehrara and Baum, 1990*). This likely serves a reproductive function as the male may use previous experiences to increase the likelihood of encounters using location preferences with an estrous female and potential mating opportunity (*Gleason et al., 2009*). Based on the knowledge of functions of T, one might predict that increased T causes males to allocate more time toward mate guarding, courting, or aggressively pursuing other males. In the current study, however, the change in spatial preference was most likely not a result of behavioral changes other than the T-induced CPPs. We found no evidence for increased mate guarding behavior since females spent more time away from the nest while males spent more time at the nest. Males were not increasing their sexual behavior (e.g. mate guarding and courtship) which would be characterized by classical rodent appetitive/courtship behavior consisting of following behavior and maintaining close proximity to their mate (*Gleason and Marler, 2010*), instead, T-pairs spent more time apart than C-pairs. Additionally, T-males did not increase USVs associated with courtship (sweeps) that unpaired males express at high levels toward unfamiliar females (sweeps; *Pultorak et al., 2015*), as would be expected from courting an unfamiliar female (although these are more difficult to detect with our field set-up). This lack of increased sexual behavior to unfamiliar females is also consistent with the finding that the administration of a single T-pulse caused paired but not unpaired male California mice to decrease sweep USVs to unfamiliar females in the laboratory (*Pultorak et al., 2015*), suggesting a dampening of the classical increase in vocalizations that occurs in response to the combined stimulus of T and the presence of a female in rodents (review by *Marler and Monari, 2021*). In the context of the nest site, there was no evidence in the current study that T-pulses increased aggression (see laboratory studies focused on male-male interactions; *Marler and Trainor, 2020*), as evidenced by lack of injuries (all animals tested were trapped post experiment with no visible injuries) or increase in aggressive barks or shortening of SV calls (*Supplementary file 1G*; see *Pultorak et al., 2018*, for evidence that barks can be produced in male-female interactions). We cannot, however, rule out that males may have been actively pushing females out of the nest as has been anecdotally observed in laboratory situations by either sex when challenged by an intruder (Rieger and Marler, unpublished data). What then were males doing at the nest? In this case, the most likely explanation is increased paternal behavior (when pups were at the nest) in the form of increased nest defense or paternal care of pups based on evidence, described below, that T can directly increase paternal care in California mice in the laboratory or possibly as a by-product of spending more time at the nest. We

suggest that T increases the focus on the reproductive or aggressive behaviors most relevant at that time depending on the social and physical contexts for that specific species (*Hurley and Kalcounis-Rueppell, 2018*). This is consistent with previous findings that the ability to create T-induced conditioned location preferences is plastic and varies with social experience and current social and physical (e.g. familiar versus unfamiliar locations) contexts (*Zhao and Marler, 2016*). Finally, we cannot rule out the alternative that males simply spent more time at the nest without altering paternal or direct pup defense behaviors. It would be valuable in the future to examine the natural expression of T-pulses in males in response to social stimuli in the field.

In nature, T-pulse release following a sexual encounter most likely occurs at the nest site (as is characteristic of rodents) when females first approach a male that has established a territory. In addition, T-pulses are expected to occur when the female is in postpartum estrus (*Gubernick and Nelson, 1989*). Therefore, T-induced CPPs could be the mechanism for increasing paternal care indirectly through increased preference for spending time at the nest. In addition, T can promote paternal care in male California mice and other species (e.g. *Juana et al., 2010*; *Luis et al., 2013*; *Ziegler et al., 2004*); although this is variable among species (review by *Hirschenhauser et al., 2003*). California mouse pups demand extensive paternal investment because they are altricial and exothermic and depend on adult presence to maintain their body temperature (*Gubernick and Alberts, 1987*). In the California mouse, the presence of the father has a significant positive effect on offspring survival when temperatures are low and the parents have to forage, but there is no effect of father's presence on pup survival when exposed to warm temperatures in the laboratory (*Gubernick et al., 1993*). The importance of the father, however, is highlighted by findings in the wild that paternal presence has a significant positive effect on offspring survival in the field (*Gubernick and Teferi, 2000*), and in laboratory studies (*Bambico et al., 2013*; *Cantoni and Brown, 1997*; *Rosenfeld et al., 2013*). The main limiting factor in California mouse reproduction is water availability (*Nelson et al., 1995*). When reproduction occurs during harsh environmental conditions and offspring require constant care, there must be a balance in the time invested toward offspring maintenance and time spent toward foraging and resource defense. To achieve balance, biparental care is essential for facilitating offspring survival and maximizing reproductive success. We, therefore, propose that in some biparental species, T-induced CPPs could be a mechanism for keeping the male at the nest to care for the young while the female forages or conducts other behaviors related to territory maintenance. Females are territorial and aggressive and also actively approach intruders or playbacks of intruders of both sexes (e.g. *Davis and Marler, 2003*; *Davis and Marler, 2004*; *Rieger and Marler, 2018*; *Rieger et al., 2019*; *Rieger et al., 2021*; *Monari et al., 2021*). Another selection pressure for T-induced paternal behavior may be increased protectiveness of pups to prevent the high levels of conspecific infanticide found in rodents (*Agrell et al., 1998*). *van Anders et al., 2012* speculate that infant protection may be positively associated with T and more nurturing behaviors negatively associated with T. In summary, the reinforcing effects of T-pulses may function to allocate more time in the familiar environment and display behaviors that have direct fitness benefits.

One possibility for why females changed their spatial preference to be away from the nest is to compensate for the T-induced changes in male spatial preferences. This is consistent with laboratory studies finding that a reduction in paternal behavior is associated with an increase in maternal huddling behavior (*Trainor and Marler, 2001*), although no compensation was found in other California mouse studies (review by *Bester-Meredith et al., 2017*). Results are varied in prairie voles as well (*Ahern et al., 2011*; *Kelly et al., 2020*). Ours is the first field study to indirectly test this idea of maternal adjustment for level of paternal care. We also observed plasticity in female but not male time at the nest in different seasons, suggesting plasticity in maternal behavior in response to environmental factors. We speculate that plasticity in the males is influenced by T from social stimuli, whereas the plasticity we see in the females may be influenced more directly by the physical environment. In species that form pair-bonds where both members of a pair are engaged in offspring care and territory defense, the delegation of tasks is beneficial. In a wider variety of taxonomic groups, including insects, birds, fish, and mammals that engage in cooperative breeding, members of a pair or group often distribute tasks (*Arnold et al., 2005*; *Ahern et al., 2011*; *Mathews, 2002*; *Page et al., 2006*; *Quinard and Cézilly, 2012*; *Rieger et al., 2019*; *Rogers, 1988*). In the laboratory, when challenged with a potential intruder, California mouse pairs either coordinate their behavior in joint defense or employ labor division strategies, with the latter strategy potentially more likely to occur after pups are

born (*Rieger et al., 2019*). In the California mouse, when the male is present but decreases paternal care due to castration, the female compensates for her mate's behavior by increasing huddling with her pups (*Trainor and Marler, 2001*). In species in which both members provide offspring care, such as in the Midas cichlid, great tit, and prairie vole, the presence of offspring increases the pairs' use of division of labor (*Rogers, 1988*; *Ahern et al., 2011*; *Boucaud et al., 2016*; *Rogers et al., 2018*; ). This division of labor can have important long-term benefits for the persistence and survival of a social group (*Arnold et al., 2005*). In the case of California mice, if the male is spending more time in one location, such as the nest to care for offspring, the female is adjusting her space use by allocating more time to other parts of the territory, such as foraging and/or defending the territory against potential intruders. These results suggest that T-pulses can alter space use and, importantly, females can adjust their behavior to compensate for male changes in space use.

## T and vocal communication

We also found that the same transient increases in T that induced CPPs also had long-term effects (>24 hr) on vocal communication by increasing the number of USVs produced and altering both the type of calls produced and the call bandwidth. T increases vocalizations in a number of species when administered as a long-term change in T (as described earlier). Our results are consistent with these other studies and *Timonin et al., 2018*, also found a nonsignificant trend for a positive effect of T-pulses on USVs in California mice in the wild. T-pairs from both studies produced proportionally more 4 SVs, demonstrating that this effect is repeatable. One difference between the studies is that *Timonin et al., 2018* found that T-pairs produced proportionately more 1-, 4-, and 5SVs, whereas we only found an effect on 4SVs. The difference between the Timonin study and the current study could be attributed to year, population densities, or a higher sample size in the current study. Anecdotally, densities were lower in the current study which could alter social interactions.

When taking into account spatial distribution, we also found that T-pairs were more likely to produce 1-, 2-, and 4SVs when >2 m (distance was not examined in *Timonin et al., 2018*). We speculate that at least 4SVs are being used to communicate between spatially separated pairs, as suggested by *Briggs and Kalcounis-Rueppell, 2011*. The current study also reveals that the increased time apart in T-pairs may indirectly drive the greater number of USVs produced by the T-pairs. However, while pairs call more when separated regardless of treatment, there was a nonsignificant trend for T to increase calling rate when pair members were >2 m apart (p = 0.09). There was also a significant treatment effect on the proportion of specific SV call types when examined specifically at >2 and 1–2 m apart. We cannot exclude a territorial function to the vocalizations, although it is important to note that these calls are being produced relatively near the nest. This study does not address what occurs when mice are even farther apart, such as one in the nest and one at the territorial boundary.

We found that the increase in SV production was associated with a decrease in bandwidth. Narrow bandwidth SVs may be more efficient for longer distance communication as narrow bandwidth USVs are less susceptible to environmental degradation and may travel further (*Barber et al., 2010*; *Slabbekoorn, 2013*; *Zhang et al., 2015*). Contrary to our findings that T-pulses decreased bandwidth in SVs, in the golden hamsters (*Mesocricetus auratus*) T-pulses increased bandwidth of calls, but these were produced in close proximity (*Fernández-Vargas, 2017*). Singing mice (*Scotinomys teguina*) administered T-implants produced mating calls also with increased bandwidth, which females tended to prefer (*Pasch et al., 2011a*; *Pasch et al., 2011b*). We speculate that under the conditions of male-female interactions in a mate-choice context, the function of the bandwidth change may be related to the increased call complexity and greater information transfer characteristic of wider bandwidths. California mice may not follow the same pattern of call production as in golden hamsters and singing mice because in our study they are likely directing SV calls toward the other member of the already established pair (*Briggs and Kalcounis-Rueppell, 2011*). Moreover, calls are unlikely to be directed toward pups because in the current study offspring presence did not influence call production. It is also possible, however, and remains untested, that the calls serve a dual function, as mate contact calls and/or as territorial advertisement. Call production most likely serves to at least maintain awareness of the other individuals in a complex environment (*Hurley and Kalcounis-Rueppell, 2018*).

We have considered the generalizability of our findings within the STRANGE framework which considers trappability, rearing, acclimation, responsiveness, genetic structure, and experience (*Webster and Rutz, 2020*). That we were working on free-living wild animals is a strength of this

contribution, in spite of relatively small sample sizes, precisely because there are no concerns regarding lab artifacts of rearing, responsiveness, acclimation, and genetic structure. In this sense our results are more generalizable than captive studies where there can be concerns about housing, rearing, inbreeding, and captivity. We sampled wild mice within a representative and historically well researched wild population over a long time frame. This leaves two issues for consideration: trappability and experience. We relied on well understood and non-attractant standard and well understood trapping methods for mice over months' long field seasons that allowed us to be sure that we had marked and were recapturing the majority of individuals who were both present and resident. This is reflected in our exceptional number of trap nights in this study. It is possible, however, that our trapping was biased toward bold or 'trapable', individuals but we know from the extensive trapping in this study, and at this site historically, that we were likely to have sampled all resident males, independent of this bias. Thus, it is likely that both trapable and less trapable animals are included in our study and the design of blind assignment of treatment means that we have both (or a continuum) in our treatment and control group. Because we were sampling resident animals from a wild population for only a few weeks during their lifetime, we could not control for differences in experience. However, the lack of information on experience is also mitigated by studying animals in the wild because it is likely that males in both our treatment and control groups were phenologically matched given that they were, at least, experienced enough to have established territories and mates.

In summary, this is the first field study that demonstrates a potentially natural function of transient T-pulses, that of inducing place preferences, possibly through CPPs. T-pulses naturally occur in a variety of different species, including humans (review by *Fuxjager et al., 2017*), and our results are consistent with other research in which T-pulses have rewarding properties and can condition animals to the physical location in which the hormone release occurred (e.g. *Arnedo et al., 2000*; *Frye et al., 2001*). We now know that despite T being weakly reinforcing compared to many drugs, it can alter behavior and do so in a complex natural environment. This change in the allocation of time spent in specific physical environments is also associated with changes in call production, likely resulting, in part, from T-induced changes in social interactions. When T altered male time spent at the nest, it may also have resulted in increased paternal behavior, and a compensatory decrease in maternal behavior. We speculate that there could be an adaptive significance for a co-option mechanism that allows a close association between mating release of T and paternal behavior. While we have effectively demonstrated potential functions of T-pulses in the laboratory and field through the current and previous studies, we do not yet know if these functions differ from those of T-implants that mimic the longer-lasting seasonal changes such as breeding versus non-breeding season (*Wingfield et al., 1999*). We speculate, however, that the T-pulses are tied in with active learning from a changing social environment during the breeding season in relation to functions related to reproduction. Once thought to be of little importance, especially in humans (*Geniole et al., 2020*), we are discovering that T-pulses have the potential to allow males to adjust to changing social conditions in the wild through both spatial preference and vocal plasticity of a male and his mate.

## Materials and methods

Field work was conducted at the Hastings Natural History Reservation (HNHR), Carmel Valley, CA, USA, from January to June 2015 (spring) and from September to December 2015 (fall) on established trapping grids. The trapping methods we used are well established and reliably capture and recapture resident mice in their territories (see details in *Briggs and Kalcounis-Rueppell, 2011*; *Kalcounis-Rüppell and Millar, 2002*; *Kalcounis-Rueppell et al., 2006*; *Kalcounis-Rueppell et al., 2010*; *Timonin et al., 2018*). Our methods include high trapping efforts to ensure a high probability of capture for all resident individuals at our study site; in this study we had 169,222 trap nights over 211 nights that include pre-experiment and experiment nights. For California mice, an average of 1500 m² territory size has been recorded (*MacMillen, 1964*); we studied their behaviors at the nest and the 2 m area immediately around the nest. Traps were set as evenly as possible around the nest based on terrain. The traps were set at sunset and checked twice per night, once at midnight and the second time around 5 AM. Of the 323 mice tagged, we identified 33 reproductively active mated pairs (males with enlarged testis and females were pregnant and/or lactating). Once putative pairs were identified, we trapped the pair and both the male and the female were outfitted with a 0.55 g M1450 mouse style transmitter (Advanced Telemetry System [ATS], Isanti, MN), adjusted for California mice (*Briggs and*

Kalcounis-Rueppell, 2011). We attached the transmitters (*Briggs and Kalcounis-Rueppell, 2011*) and

released all mice at the site of capture. Using an R4500S DCC receiver/datalogger and a Yagi antenna
(ATS), we located the pair the following day at the nest (described below). All 33 putative pairs were
confirmed as pairs when the signals from both the male and female transmitters were emitted from
the same nest. We ensured that the tracked nest location was the primary nest and not one of the
satellite locations by monitoring nest occupancy for up to 3 days. A total of 28 pairs (reduced to 27
because of telemetry issues) were in the nest for up to 3 days post tracking, and we ensured that the
nest was in a suitable location for setting up our remote sensing equipment (described below).

## Treatment

We randomly assigned 28 males to receive either T (n = 15) or saline (control, C, n = 13) injections.
Sixteen traps were placed within a 2 m radius around the nest, such that the nest was in the middle.
The focal male was removed from the trap, injected and immediately released at the opening to the
nest. The male would then retreat to the nest. For the following treatments, we recaptured males
three times, on three subsequent nights, within 2 m of the nest. All traps were set at sunset and
checked twice per night, once at midnight and the second time around 5 AM. The dose of T-injection
was approximately 36 µg/kg (T-cyclodextrin dissolved in saline) which mimics natural T-pulses (*Oyeg-
bile and Marler, 2005*; *Trainor et al., 2004*) and has been used successfully in multiple California
mouse studies primarily focused on aggression and courtship (*Fuxjager et al., 2011*; *Pultorak et al.,
2015*; *Timonin et al., 2018*; *Trainor et al., 2004*; *Zhao and Marler, 2014*; *Zhao et al., 2020*; *Zhao
et al., 2019*). Prior to injection administration, the health of each individual was assessed using the
grimace scale. All animals were restrained by the scruff of the neck and the needle was inserted at
the base of the fold between the researcher's fingers to administer the injection subcutaneously, and
the researcher was blind to the treatment type. Each focal male received three injections of 0.1 ml
of the injectate regardless of body mass, with only one injection on any given night. We, therefore,
included body mass as an independent variable in our statistical analysis. All three injections were
administered within five nights. One male was excluded because he did not receive all three injections
within 5 days. We refer to females whose mate received T as 'T-females' and the nests as 'T-nests'.
Females whose mate received saline are referred to as 'C-females' and their nests as 'C-nests'. We
also recorded the total number of nights needed to administer all three injections (three or four
nights), and included total nights as an independent variable in our statistical analysis. After the third
and last injection, we deployed the remote sensing equipment (automated radio telemetry, audio
recording, and thermal imaging; described below) to record for three consecutive nights ('recordings
nights' 1–3). We treated data collected by the remote sensing equipment over one night as a sample
unit and included recording night in our analyses. For each recording session, all equipment was set
up to record from sunset to sunrise. T and C solutions were provided by Dr Brian Trainor from the
Department of Psychology at the University of California Davis (IACUC Protocol number 19849).

## Automated radio telemetry

We used two R4500S DCC receiver/dataloggers (ATS, Isanti, MN) to monitor the number of minutes
radio-collared mice spent at the nest each night and the amount of time the male and female were
together and apart. Each data logger was connected to an antenna and programmed to detect
one unique transmitter frequency per pair member. Antennas were placed either on top of or next
to the nest. When the collared mouse was detected by the receiver, signal strength was stored in
the datalogger, we could therefore frequently track male and female movements separately. We,
therefore, monitored both male behavioral changes in response to treatment type and the female
response to male behavioral changes. Because there were differences in length of recordings due
to differences in length of night such as by season, we standardized the time at the nest. We first
counted the number of minutes the mouse spent in the nest and then divided by the duration of
the night (total minutes from sunset to sunrise). We were able to measure male time at the nest
and female time at the nest, separately and together. We do not know where on the territory the
animals were spending the time when they were away from the nest because we focused our moni-
toring on the nest. Each day we also conducted manual telemetry on the collared pair and found
the nest location with the strongest signal strength. For each individual, we assessed a reference
signal (range 130–155 dB signal strength) during the day when we knew the mouse was in the

nest. To assess how long a mouse spent in the nest and the 2 m area around the nest per night, we only counted the number of minutes during which the signal fell within the reference range. Each morning, the data loggers were removed from the field and data were downloaded. The telemetry equipment was set up at 27 nest sites. Due to equipment failure, we did not record male time at the nest for five T-nests and one C-nest and we did not record female time at the nest for one T-nest and three C-nests. Our final dataset consisted of 63 recording nights from 21 nest sites (T = 10, C = 11) for males and 69 recording nights from 23 nest sites (T = 14, C = 9) for females. We did not have matching pair time at the nest for five T-nests and four C-nests. Our final matching pair dataset consisted of 54 recording nights from 18 nest sites (T = 10, C = 8) and we used night as a sample unit in our analysis.

## Audio recording

Our goal was to record all the different types of USVs. The SVs have a peak frequency of around 20 kHz, and are approximately 50–1000 ms in length; these are low modulation calls that can be emitted as a single or bout of multiple calls that can be categorized based on the number of calls in a bout (1-, 2-, 3-, 4SV, etc.; *Kalcounis-Rueppell et al., 2018*). Bark calls are shorter in duration (50 ms or less), resemble an upside-down U with the beginning and the end of the call dips into audible range at approximately 12 kHz with a peak frequency around 20 kHz and tend to be 'noisy' vocalizations (*Pultorak et al., 2018*). Similar to the SVs, the barks occur as a single call or bout of calls.

We used ultrasonic microphones (Emkay FG Series from Avisoft Bioacoustics, Berlin, Germany) to assess the number and type of USVs produced at the nest. We set up two microphones: one next to the nest entrance and a second 2 m away directly from the nest entrance. Microphones recorded as described in *Timonin et al., 2018*. When possible, we assigned USVs to individuals by matching the radio telemetry data with the time of the mouse USV. By examining telemetry data within 1 min of USV production and based on the transmitter signal strength (*Briggs and Kalcounis-Rueppell, 2011*), we determined if the male or the female produced the USV. We were not able to assign 51% of the USVs to one individual because both the male and the female were at the nest with strong transmitter signal strengths and therefore, we only used the assigned data to test the treatment effect on the spectral and temporal characteristics of USVs. The acoustic recording system was set up at 27 nest sites (T = 15, C = 12). Due to equipment failure, we did not record data at one T-nest. Our final dataset consisted of 78 recording nights from 26 nest sites (T = 14, C = 12). Mouse USVs were counted and classified into one of the following types: 1-, 2-, 3-, 4-, 5-, 6SVs, or barks (*Kalcounis-Rueppell et al., 2018*). We counted USV numbers recorded from sunset to sunrise and refer to the value as 'total USVs'. Lastly, we determined if the proportion of a specific type of USV (1-, 2-, 3-, 4-, 5-, 6SVs, and barks) differed between treatments by totaling each USV type per nest site and dividing by the total number of USVs produced at that nest.

Using SAS Lab Pro, we extracted spectral and temporal characteristics from calls recorded at the nest. Each spectrogram was generated with a 512 FFT (fast Fourier transform), and a 100-frame size with a Hamming window. For each call, we measured duration, bandwidth, and five frequency parameters (start, end, minimum, maximum, and frequency at maximum amplitude).

## Thermal imaging

We used a thermal imaging lens (Photon 320 14.25 mm; Flir/Core By Indigo) to assign social context to USVs. The thermal imaging lens was suspended to capture the full view of the nest and a circular area with a 2 m radius surrounding the nest. The lens was connected to a JVC Everio HDD camcorder which recorded continuously throughout the night. We watched the video footage in 3 min increments (1 min before, 1 min during, and 1 min after call production) to determine behavior and number of mice on the screen. If both mates were present, we determined the proximity of mice to each other by using a 1 m scale that was overlaid in the video for each site. If mice were less than 1 m apart, we assigned them as '<1 m', and if the mice were more than 1 m apart, we marked them as '1–2 m'. If there was only one member of a pair present at a time, the behavior was assigned as >2 m. We assessed the types of USVs (1-, 2-, 3-, 4-, 5-, 6SVs, and barks) produced by context (<1, 1–2, or >2 m) and treatment type.

## Statistical analyses

Time at the nest for both the male and the female was normally distributed and therefore we fitted a Gaussian distribution. Pair time at the nest and total USVs were in violation of normality and variances and could not be normalized and therefore we used either a quasibinomial and/or Poisson distribution, respectively. We used generalized linear mixed models (GLMMs) with time at the nest, pair time at the nest, and total USVs as the dependent variables and included individual identification code (ID) as a random term, independent of treatment type to account for individual differences. Using the package lme4 (*Bates et al., 2015*), we fitted a repeated measure GLMMs with ID as a random term and treatment as the fixed term.

In addition to treatment type, we also considered the following covariates: presence of pups at the nest, season, male, and female body mass, total nights needed to administer all three injections, and recording night. Due to our small sample size, when modeling covariates we included a maximum of two fixed terms in one GLMM (treatment type and one covariate per analysis). We first modeled the interaction term between treatment type and the one covariate. If the interaction term was not significant, the term was dropped. We also used the non-parametric Wilcoxon rank sum test for our comparison of USV types. We compared the median of the proportion of each USV type by treatment. We performed GLMs to examine the relationship between all USVs combined and distances from the nest as described above under thermal imaging. We performed the chi-squared test of independence to examine if there was a relationship between specific USV types and distance from the nest. For the analysis of the spectral and temporal characteristics, we used factor analysis to extract principal component (PC) scores for the frequency parameters (as in *Kalcounis-Rueppell et al., 2010*). For this analysis, we only analyzed calls assigned to an individual male or female and the calls were analyzed separately. We generated a single PC score that represented the frequency variables using the first call in the 1-, 2-, 3-, and 4SVs sequence. We did not include 5-, 6SVs, and barks due to a small sample size (<4), however, the numbers are reported in *Supplementary file 1E*. PC1 accounted for 67% of the variation in frequency variables for male calls and 71% variation for female calls (*Figure 5—figure supplement 1*). Our dependent variables were PC1, call duration, and call bandwidth. We fitted GLMM with ID as a random term and USV type and treatment as the fixed terms. For both male and female calls, duration and bandwidth variables were in violation of normality and variances. We, therefore, fitted our models using a Poisson family distribution. PC scores were normally distributed, and we used a Gaussian distribution in our models. All data are represented using box plots. Our data are analyzed as repeated measures and this is represented in the text and figures, however, we also added an analysis whereby we averaged the three nights and there is no loss of statistical significance using this method (Appendix 1). We used an alpha level of $p < 0.05$ for the rejection criterion. All data were analyzed using R software (Version 3.2.2.)

## Acknowledgements

We thank A Campos, J Caprio, C Falvo, M Grupper, C Kovarik, A Larsen, J Neill, and E Sakonjic for their assistance in data collection and Brian Trainor for providing our C and T injectate. We also thank J del Valle, V Voegeli, and Hastings Natural History Reserve for their support and use of their facilities. Lastly, we appreciate the input from H Li on all aspects of the analysis and B Trainor, R Bhandari, G Wasserberg, and C Snowdon for comments on the manuscript. This work was supported by the National Science Foundation (NSF; IOS-1355163) and UNC-Greensboro.

## Additional information

### Funding

| Funder | Grant reference number | Author |
| --- | --- | --- |
| National Science Foundation | 1355163 | Matina Kalcounis-Rueppell Catherine A Marler |
| Sigma Xi | Spring 2018 | Radmila Petric |

| Funder | Grant reference number | Author |
|---|---|---|

The funders had no role in study design, data collection and interpretation, or the decision to submit the work for publication.

## Author contributions

Radmila Petric, Conceptualization, Formal analysis, Investigation, Methodology, Project administration, Visualization, Writing – original draft; Matina Kalcounis-Rueppell, Conceptualization, Funding acquisition, Resources, Supervision, Writing – review and editing, Project administration; Catherine A Marler, Conceptualization, Funding acquisition, Resources, Supervision, Writing – review and editing

### Author ORCIDs
Radmila Petric ⓘ http://orcid.org/0000-0002-2651-3328
Matina Kalcounis-Rueppell ⓘ http://orcid.org/0000-0003-0964-2125
Catherine A Marler ⓘ http://orcid.org/0000-0002-2783-1029

### Ethics

All animal care and use guidelines were followed and research protocols for this study were approved by the University of North Carolina at Greensboro and University of Wisconsin-Madison Institutional Animal Care and Use Committees (IACUC; UNCG 12-004 and UWM L005047-A01) and by California Department of Fish and Wildlife under Scientific Collection Permits (SC-9663 and SC-13190).

### Decision letter and Author response
Decision letter https://doi.org/10.7554/eLife.65820.sa1
Author response https://doi.org/10.7554/eLife.65820.sa2

## Additional files

### Supplementary files
• Supplementary file 1. Male time at the nest.
• Transparent reporting form
• Source data 1. Source data for male and female time at the nest.
• Source data 2. Source data for call production.

### Data availability

All data analysed for this study are included in the manuscript and supporting files. Source data files have been provided for all figures, and are available on OSF (https://doi.org/10.17605/OSF.IO/QKNZE).

The following dataset was generated:

| Author(s) | Year | Dataset title | Dataset URL | Database and Identifier |
|---|---|---|---|---|
| Petric R | 2021 | T-pulses at the Nest | https://doi.org/10.17605/OSF.IO/QKNZE | Open Science Framework, 10.17605/OSF.IO/QKNZE |

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

## Appendix 1

### Statistical analysis conducted with average proportion of time spend at the nest

T-males spent 14% more time at the nest than C-males (GLMM Estimate 0.13±0.05, p=0.03). Males and females spent more time at the nest when there were pups (male time at the nest and pups GLMM Estimate 0.20±0.03, p<0.00; female time at the nest and pups GLMM Estimate 0.16±0.06, p<0.02), however, sample sizes were too small to statistically compare both pup presence and treatment type in one model. Male time at the nest was not statistically influenced by season (GLMM Estimate 0.07±0.06, p=0.26), body mass (GLMM Estimate −0.01±0.01, p=0.47), and total nights needed to administer all three injections (GLMM Estimate −0.16±0.08, p=0.06). Females were not subjected to T-injections, but we examined their responses to their T-injected mates. T-females spent 17% less time at the nest than C-females (GLMM Estimate −0.16±0.07, p=0.02). T-females spent 15.2% less time in the nest during spring than fall (spring GLMM Estimate −0.15±0.07, p=0.04). Female time at the nest was not statistically influenced by body mass (GLMM Estimate −0.01±0.01, p=0.27).

