## [Editor Report]

Manipulations of sex hormones in animals in ecologically relevant environments usually involve long-term manipulations using chronic implants or injections of esterified steroids with longer half-lives than the endogenous hormones. This has been done in line with the prevailing idea of the long-lasting effects of steroids mediated by the transcritpional actions of their liganded receptors. The specific novelty of this study lies in the transiency of hormone availability (testosterone's half-life is about 2 hours). This might suggest that the observed effects depend on a mode of action different from the mode of action during chronic sex hormone exposure. It should also be noted that any study in natural settings is significantly more difficult to perform than in the lab. However, as all brain/hormonal functions evolved in natural environments, these studies are absolutely crucial to understanding the function of the respective systems.

---

## [Decision Letter]

**Decision letter after peer review:**

Thank you for submitting your article "Testosterone induces a conditioned place preference to the nest of a monogamous mouse under field conditions" for consideration by *eLife*. Your article has been reviewed by three peer reviewers, and the evaluation has been overseen by a Reviewing Editor and Christian Rutz as the Senior Editor. The reviewers have opted to remain anonymous.

The reviewers have discussed their reviews with one another, and the Reviewing Editor has drafted this decision letter to help you prepare a revised submission. The reviewers concluded that several major and minor points must be addressed to make a compelling case for publication in *eLife*.

Essential revisions:

1) Please address more clearly the novelty of the study. This is very important, as there was a discussion among the reviewers and reviewing editor whether or not the manuscript is too narrow in scope and offers too little novel insight compared to existing literature. Please pay particular attention to the transiency of the hormone availability and the notion that observed effects would depend on a different mode of action of the steroid, i.e. that the reinforcing properties of testosterone are mediated by a rapid and transient mechanism presumably involving membrane-initiated signaling rather than by the transcriptional activity of its nuclear receptor.

2) While it is clear that testing large numbers of animals in an experiment such as this is not trivial, the sample sizes are too small to support many of the results, especially when you attempt to estimate the effect of so many covariates. For example, there are only 4 male mice injected with testosterone that have pups. Fitting so many variables when overall sample size is small is problematic and can lead to false positive results. The study should focus on the variables for which they have sufficient power.

3) In the revised version of the manuscript, please critically discuss the following two concerns:

– While you find that mice injected with testosterone spend more time at the nest, it is not clear that the effect is specific to the nest, or if testosterone would induce a conditioned place preference to other spatial locations as well. Moreover, it is not clear from these analyses if animals are spending more time at the nest simply because testosterone made them sick, for example.

– What is the evidence that the experimental manipulations are mimicking natural testosterone pulses at the nest? Would other rewards or pharmacological agents known to produce conditioned place preference, administered at the site of the nest, produce a different behavioral profile than what was observed with testosterone?

4) Evaluate USV data in animals with and without pups who are administered with control vs. testosterone injections.

[Editors' note: further revisions were suggested prior to acceptance, as described below.]

Thank you for resubmitting your work entitled "Testosterone pulses paired with a location induce a place preference to the nest of a monogamous mouse under field conditions" for further consideration by *eLife*. Your revised article has been evaluated by Kristin Tessmar-Raible (Reviewing Editor) and Christian Rutz (Senior Editor).

We apologize that it took us so long to come to a conclusion about your revised manuscript. Unfortunately, of the three original reviewers, one did not have the time to re-evaluate your manuscript and the other two disagreed on the general interest level of your study, as well as the sufficiency of your revisions. For the general interest level, we obtained an independent opinion, which took a while, but was very positive. However, there are multiple aspects in your revision that at present preclude acceptance for publication.

Most importantly, several points suggest that not enough care was taken during the preparation of the revised manuscript and the cover letter. This includes the following points:

– You refer to lines where you incorporated changes, yet these lines don't exist in the paper (e.g., lines 1172-1175) or refer to the wrong section (e.g., line 420, lines 970-973).

– You changed the main Figure 2 to "Proportion of time spent at the nest" as suggested by the reviewers, but did not make this change in the supplementary Figure 2.

– In the response letter, you refer to SI Figures 2 and 3, but there is no SI Figure 3. The labeling of the figures in the manuscript file, cover letter and in the *eLife* upload is inconsistent.

– Where is Figure 1 in the document with the tracked changes?

All of these issues can be fixed, but raise concerns that insufficient care was given to other aspects of the work. Please check all materials again very carefully to bring them up to a standard that makes the manuscript acceptable for publication in *eLife*.

Furthermore, it will also be a prerequisite for publication that ALL raw data will be made available, including the call files. Please also ensure that the Open Science Framework doi under which you provide the data can be found online. We checked your response to Sam Porteous concerning this aspect. While the link is "clickable", the copy/paste of the doi does not seem to work, suggesting that it is not fully openly available. (This caused the problem that the Reviewing Editor accidentally downloaded another manuscript's datafiles, which then obviously did not fit with your manuscript.)

There are still major concerns about the definition of replicates. Please state clearly in the figure captions that the dots currently represented in the figures are the number of observations. Please also provide additional analyses in which 'n' represents the number of animals (with the average of observations/animal and condition as a single number), and relevant statistics. In case this results in loss of statistical significance, please flag this for editorial attention in your response letter and also discuss any implications in your revised manuscript.

Finally, please note that *eLife* has recently adopted the STRANGE framework, to help improve reporting standards and reproducibility in animal behaviour research. In your final revision, please consider scope for sampling biases and potential limitations to the generalisability of your findings.

---

## [Author Response]

Essential revisions:1) Please address more clearly the novelty of the study. This is very important, as there was a discussion among the reviewers and reviewing editor whether or not the manuscript is too narrow in scope and offers too little novel insight compared to existing literature. Please pay particular attention to the transiency of the hormone availability and the notion that observed effects would depend on a different mode of action of the steroid, i.e. that the reinforcing properties of testosterone are mediated by a rapid and transient mechanism presumably involving membrane-initiated signaling rather than by the transcriptional activity of its nuclear receptor.

Thank you for this suggestion. We have added more information to the Introduction which should address this comment by contrasting with T-implant studies more extensively and emphasizing nongenomic effects of T to a greater extent (page 2, lines 161-168). It is also important to understand that this is the first time that the concept and potential function of conditioned place preferences has been addressed under natural conditions. While it is not conducted as a classical conditioning study, this is not possible in the wild. There will always be additional stimuli and that means that the natural function of CPPs are likely broader than ideas that have been derived from the laboratory paradigms. Context, both social and spatial, will likely turn out to be critical for expression of such behaviors. Associative learning created by T release at a specific location results in an end product that includes a place preference in addition to other possible associations. Clearly further studies are needed, but this study provides the first critical step and does so in a natural environment.

2) While it is clear that testing large numbers of animals in an experiment such as this is not trivial, the sample sizes are too small to support many of the results, especially when you attempt to estimate the effect of so many covariates. For example, there are only 4 male mice injected with testosterone that have pups. Fitting so many variables when overall sample size is small is problematic and can lead to false positive results. The study should focus on the variables for which they have sufficient power.

We acknowledge that the sample sizes are small, however, we feel strongly that these results are important to present and in this revision all of our groups have a minimum number of 8 and a maximum number of 14 individuals. Given these sample sizes the statistics we conducted are appropriate. We also acknowledge the especially small sample size associated with pups and have therefore removed the pup results from the main text as suggested and this can now be found in the Supplementary Information (Figure 2 —figure supplement 1 and Figure 2 —figure supplement 2; Supplementary file 1A and Supplementary file 1B).

3) In the revised version of the manuscript, please critically discuss the following two concerns:– While you find that mice injected with testosterone spend more time at the nest, it is not clear that the effect is specific to the nest, or if testosterone would induce a conditioned place preference to other spatial locations as well. Moreover, it is not clear from these analyses if animals are spending more time at the nest simply because testosterone made them sick, for example.

We thank the reviewer for pointing out that we should address this issue in the paper. The first author has extensive experience handling and assessing overall health of mice in the wild using the grimace scale. In our study design, males received 3 injections over the course of 5 days which means we recaptured individuals after the 1st and 2nd injection and assessed their health each time. This text has been added on page 23, lines 2908-2909. In the wild, there were no differences in behavior while handling the animals between males that received testosterone or saline as neither group showed signs of illness. Furthermore, similar methods to this study on the same species were conducted in many laboratory studies in the Marler lab under supervised conditions by laboratory technicians, graduate students and a veterinarian and there were no signs of distress or illness in those individuals either. Moreover, other studies outside of the laboratory have used both T injections and implants with no ill effects. Overall, it is highly unlikely that the testosterone made the mice sick. In addition, these doses mimic natural levels in the male P. californicus and were therefore not expected to have toxic effects as referenced in the text.

The injections were only administered at the nest site for the current study. The question regarding the ability of T to produce CPPs to other locations is very interesting. We are conducting a series of studies and are currently asking whether the same result would occur when the injections are made at the border of the territory. Preliminary results indicate that males are moving farther away from the nest when injected at the territory border.

– What is the evidence that the experimental manipulations are mimicking natural testosterone pulses at the nest? Would other rewards or pharmacological agents known to produce conditioned place preference, administered at the site of the nest, produce a different behavioral profile than what was observed with testosterone?

We have previously published a timeline of the natural changes of T in response to a male-male encounter (Marler, C.A., Oyegbile, T., Plavicki, J., and Trainor, B. 2005; Response to “A continuing saga: The role of testosterone in aggression.” Hormones and Behavior, 48, 256-258). This was measured again in Oyegbile and Marler 2005 at 45 min after the encounter (Oyegbile TO, and CA Marler. 2005; Winning Fights Elevates Testosterone Levels in California Mice and Enhances Future Ability to Win Fights. Hormones and Behavior 48 (3): 259–67. https://doi.org/10.1016/j.yhbeh.2005.04.007). The dose we selected for this study is biologically relevant based on the findings of these previous studies and the additional research from Trainor et al. 2004. This dose has been used in multiple other studies (e.g.Trainor et al. 2004; Fuxjager and Marler 2012; Zhao and Marler 2014,2016; Fuxjager et al. 2011a,b; Pultorak et al. 2015; Zhao et al. 2019) including another study in the field by Timonin et al. 2018.

Finally, we have unpublished data showing that the same level T pulse is produced by males after a male-female encounter, see Author response image 1.

**Author response image 1. sa2fig1:** Time course for plasma T levels of individual male P. californicus exposed to a female (no repeated measures). Xin Zhao and Catherine Marler, unpublished data. * p<0.05.

We cannot answer the question about whether other “rewards or pharmacological agents known to produce conditioned place preference, administered at the site of the nest, produce a different behavioral profile than what was observed with testosterone?”, although it is interesting to speculate. We would predict that the place preference would still occur. The behaviors associated with that place preference might differ, but this would require a separate set of studies based on the function of those particular pharmacological agents.

4) Evaluate USV data in animals with and without pups who are administered with control vs. testosterone injections.

USV data were evaluated in mice with and without pups and by treatment type. Our USV data results show that pup presence does not account for any of the variation in call characteristics as presented on page 8, lines 922-923. For all analyses the p-value was greater than 0.05.

[Editors' note: further revisions were suggested prior to acceptance, as described below.]Most importantly, several points suggest that not enough care was taken during the preparation of the revised manuscript and the cover letter. This includes the following points:– You refer to lines where you incorporated changes, yet these lines don't exist in the paper (e.g., lines 1172-1175) or refer to the wrong section (e.g., line 420, lines 970-973).

We corrected the original responses and they now correspond to the current pages and lines.

– You changed the main Figure 2 to "Proportion of time spent at the nest" as suggested by the reviewers, but did not make this change in the supplementary Figure 2.

The change has been incorporated into the supplementary Figure 2 as well.

– In the response letter, you refer to SI Figures 2 and 3, but there is no SI Figure 3. The labeling of the figures in the manuscript file, cover letter and in the eLife upload is inconsistent.

We have taken extra care to make sure all of the figures are referenced correctly.

– Where is Figure 1 in the document with the tracked changes?

Figure 1 is referenced on page 5, line 711.

All of these issues can be fixed, but raise concerns that insufficient care was given to other aspects of the work. Please check all materials again very carefully to bring them up to a standard that makes the manuscript acceptable for publication in eLife.

Please note that we added several changes to help the flow and clarity of the manuscript.

The manuscript has been reviewed again thoroughly to make sure that:

a) The letter and manuscript match.

b) Writing is clear, especially with figure and table legends.

c) Details are clear such as those with statistics.

d) The Open Science Frameworks doi is functional.

e) Replicates are repeated measures. We have made this clear throughout the manuscript, particularly in the figure and table legends (see below).

f) We added a full paragraph related to the STRANGE framework (see below).

Furthermore, it will also be a prerequisite for publication that ALL raw data will be made available, including the call files. Please also ensure that the Open Science Framework doi under which you provide the data can be found online. We checked your response to Sam Porteous concerning this aspect. While the link is "clickable", the copy/paste of the doi does not seem to work, suggesting that it is not fully openly available. (This caused the problem that the Reviewing Editor accidentally downloaded another manuscript's datafiles, which then obviously did not fit with your manuscript.)

All the raw call files are now included and the doi link should work either as a clickable link or when copied/pasted. Along with the raw acoustics files we have included a word document to provide a detailed description about the file organization. doi.org/10.17605/OSF.IO/QKNZE

There are still major concerns about the definition of replicates. Please state clearly in the figure captions that the dots currently represented in the figures are the number of observations. Please also provide additional analyses in which 'n' represents the number of animals (with the average of observations/animal and condition as a single number), and relevant statistics. In case this results in loss of statistical significance, please flag this for editorial attention in your response letter and also discuss any implications in your revised manuscript.

This concern has been dealt with explicitly and there is no loss of statistical significance to report (See Appendix 1). To address this concern in the manuscript, we included the following information. First, we emphasized that our statistical models used repeated measures on page 28, lines 3140. Second, we redid all of the analyses that were in our repeated measures frameworks and instead averaged the three nights for each individual. We report the outcome of this additional analysis in the statistical analysis section of the Methods on page 29, lines 3192-3195 and put the details of the additional analysis into Appendix 1. We also reminded the reader about repeated measures and referenced the additional analysis in the legend of Figure 2, Figure 2 supplement 1, and Figure 2 supplement 2. This addresses the concern because we have now clarified what “n” is representing throughout the manuscript. To make clear what the dots on the figures represent we have also added the phrase “A single dot represents the observations from one individual on a single night. Therefore for each individual there are three dots in the figure representing three nights” to the figure legends of Figure 2 and Figure 2 supplements.

Finally, please note that eLife has recently adopted the STRANGE framework, to help improve reporting standards and reproducibility in animal behaviour research. In your final revision, please consider scope for sampling biases and potential limitations to the generalisability of your findings.

Thank you for pointing this out. We agree that incorporating the STRANGE framework is important and we have added a paragraph to our discussion that considers the scope of sampling and the generalisability of our findings. This is on pages 20-21, Lines 2699-2762. This addition also included adding a sentence about our sampling effort in the methods on page 22, Lines 2826-2828.